# Chronic Nonbacterial Osteomyelitis of the Jaw in a 3-Year-Old Girl

Shigeru Makino [1], Hideo Oshige [2], Jun Shinozuka [1] and Shinsaku Imashuku [1,3,*]

[1] Department of Pediatrics, Uji-Tokushukai Medical Center, Kyoto 611-0041, Japan
[2] Department of Dentistry, Uji-Tokushukai Medical Center, Kyoto 611-0041, Japan
[3] Department of Laboratory Medicine, Uji-Tokushukai Medical Center, Kyoto 611-0041, Japan
[*] Correspondence: shinim95@mbox.kyoto-inet.or.jp; Tel.: +81-774-20-1111; Fax: +81-774-20-2336

**Abstract:** Differential diagnosis of bacterial osteomyelitis (BOM) and chronic nonbacterial osteomyelitis (CNO) is challenging. Pediatric CNO can be diagnosed at around 10 years of age and when CNO cases involve only the jaw, it is difficult to make a diagnosis in a young child. A 3-year-old female developed CNO at the jaw alone. She presented with no fever, right jaw pain, mild trismus, and a preauricular facial swelling around the right mandible. Computed tomography (CT) revealed a hyperostotic right mandible, with osteolytic and sclerotic changes associated with periosteal reaction. At first, we suspected BOM and antibiotics were administered. Subsequently, CNO was diagnosed, and the patient received flurbiprofen (a nonsteroidal anti-inflammatory drug (NSAIDs)). Lack of a sufficient response led to successful treatment with a combination of oral alendronate and flurbiprofen. Physicians should be aware of CNO, a rare autoinflammatory noninfectious bone disease of unknown etiology, even in young children, although the disease mostly affects older children and adolescents.

**Keywords:** chronic nonbacterial osteomyelitis; bacterial osteomyelitis; jaw; flurbiprofen; alendronate

## 1. Introduction

Chronic nonbacterial osteomyelitis (CNO) or chronic recurrent multifocal osteomyelitis (CRMO), i.e., CNO/CRMO, is an autoinflammatory bone disease characterized by the insidious onset of bone pain and local swelling, which usually occurs in older children, adolescents, and adults. The disease mostly affects the long bones, as well as the clavicle, pelvic bone, mandible, and spine [1–3]. A related disease is SAPHO (synovitis, acne, pustulosis, hyperostosis, and osteitis) syndrome, which involves the skin and synovia in addition to bones [4]. The pathogenesis of CNO and SAPHO may be driven by cutibacterium (formerly propionibacterium) acnes [5]. Thus, the differential between bacterial osteomyelitis (BO or BOM) and nonbacterial osteitis (NBO) is a challenge [6]. Additionally, the development of CNO may be associated with skin infection or mucocutaneous disease [7]. The involvement of the mandible in CNO is usually either isolated or occurs alongside multiple bone diseases [8–11]. In cases of CNO which involve only the jaw in a young child, it is often difficult to make a diagnosis. Here, we report a 3-year-old female who developed CNO only in the jaw.

## 2. Case Report

The patient, a 3-year-old female, visited the dentistry and pediatric clinics in the Uji-Tokushukai Medical Center, Uji, Japan. She had no fever, right jaw pain, mild trismus, and preauricular facial swelling around the right mandible, but had neither acne nor pustulosis. She was the only child in her family, and her family history was unremarkable. She had healthy dentition and no history of trauma or recurrent aphthous ulcer. Computed tomography (CT) revealed a hyperostotic right mandible, with osteolytic and sclerotic changes associated with periosteal reactions (Figure 1A), which led us first to suspect

bacterial osteomyelitis (BOM), although she did not have high C-reactive protein (CRP) values with normal white blood cell counts, and her blood culture was negative. Serum lactate dehydrogenase (LDH) and uric acid were within the normal ranges, while alkaline phosphatase was slightly elevated. Vitamin C levels were not examined (Table 1).

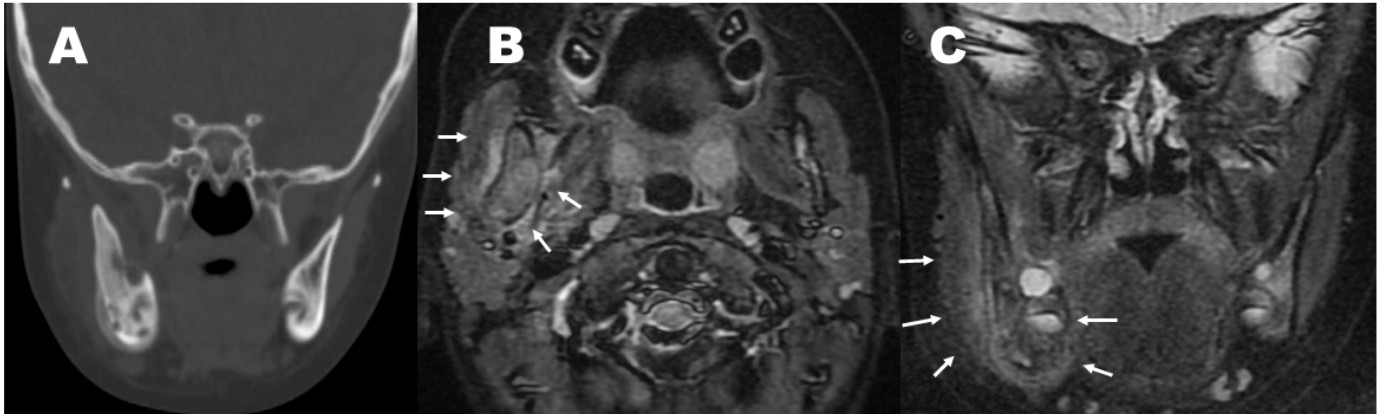

**Figure 1.** Computed tomography ((**A**); coronal view) revealed a hyperostotic right mandible, with osteolytic and sclerotic changes associated with periosteal reactions. Magnetic resonance imaging (MRI; short tau inversion recovery (STIR); (**B**), axial view; and (**C**), coronal view) showed a low-dense hyperostotic right mandible, associated with an inflamed right masseter muscle and wing pterygoid muscle (arrows).

**Table 1.** Laboratory data.

| CBC | | Inflammatory Markers | |
|---|---|---|---|
| WBC (3300–8600/μL) | 6300 | CRP (<0.14) mg/dL | 0.91 |
| Hb (11.6–14.8 g/dL) | 10.7 | Procalcitonin (<0.4) ng/mL | NT |
| Platelet counts (158 K–348 K/μL) | 405 K | ESR (1 h; 3–15) mm | 42 |
| Hepatic function | | ESR (2 h; NA) mm | 74 |
| AST (13–30) U/L | 23 | Renal function | |
| ALT (10–42) U/L | 10 | BUN (8.0–20) mg/dL | 13.1 |
| LDH (124–222) U/L | 200 | Creatinine (0.65–1.07) mg/dL | 0.24 |
| ALP (38–113) U/L | 284 | Uric acid (3.7–7.8) mg/dL | 3.0 |
| Total protein (6.6–8.1) g/dL | 7.2 | Other (vitamin C) | NT |
| Albumin (4.1–5.1) g/dL | 3.9 | Bacterial study | |
| Immunological | | Blood culture | neg |
| IgE (RIST) (<30) IU/mL | 223 | Urine culture | NT |

Abbreviations: CBC = complete blood count; WBC = white blood count; Hb = hemoglobin; AST = aspartate aminotransferase; ALT = alanine aminotransferase; LDH = lactate dehydrogenase; ALP = alkaline phosphatase; RIST = radioimmunosorbent test; CRP = C-reactive protein; ESR = erythrocyte sedimentation rate; BUN = blood urea nitrogen; NT = not tested; and NA = not available.

Physically, there was no apparent abscess or fistula at the jaw lesion. Though she received oral antibiotics (AMPC) for 1 month, her symptoms did not improve. From CT images, Langerhans' cell histiocytosis was thought to be improbable, rather, CNO was highly likely. Consultation with otolaryngologists led to some debate about whether a biopsy of the affected mandible was required to reach a correct diagnosis, as reported by several authors [7,9–11]; however, we chose no invasive measures. During the following 5 months, no clinical features of BOM manifested, and the follow-up of blood tests remained normal. At 6 months after the initial visit, MRI (STIR image) revealed a low-density

hyperostotic right mandible. In addition, inflammation had spread to the right masseter muscle, and to the right interior and exterior wing pterygoid muscles (Figure 1B,C), which was suggestive of CNO rather than BOM.

Under a probable diagnosis of CNO, the patient received flurbiprofen (Froben; 3 mg/kg/day), which is a nonsteroidal anti-inflammatory drug (NSAID). During the 6 months of treatment with flurbiprofen, she was doing well; the facial swelling was ameliorated, with only occasional mild pain in the involved jaw. However, to obtain a better outcome, we then switched treatment after 12 months; the patient received a combined treatment with oral alendronate jelly (2/5 adult dose; 14 mg; 0.7 mg/kg/day), which is normally used for adults at 35 mg/day, plus flurbiprofen (see clinical course in Figure 2). This alendronate treatment was administered once per week. After 1 month of treatment with this regimen, her symptoms subsided markedly, with normalized serum CRP levels and no acceleration of the erythrocyte sedimentation rate. At the age of 4.5 years (18 months from the initial visit), the patient was almost symptom-free. Thereafter, both drugs were tapered.

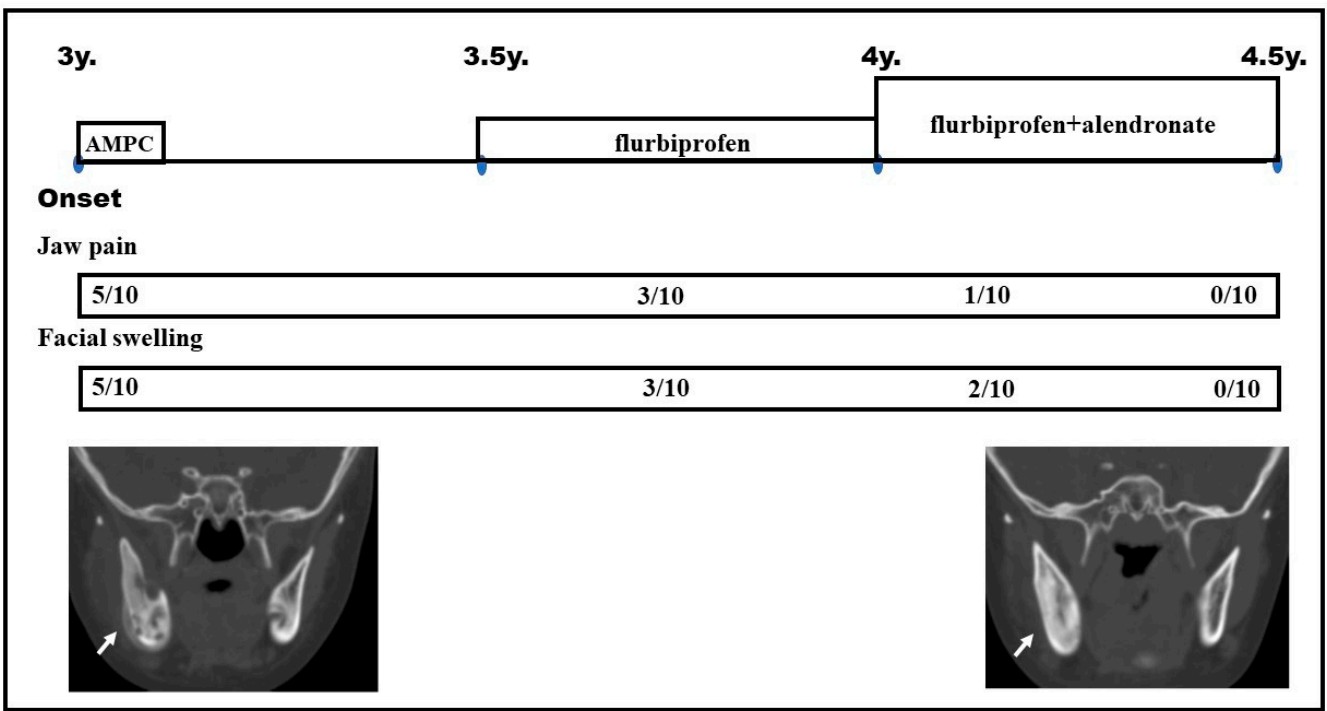

**Figure 2.** Clinical course of the patient. Antibiotics (AMPC for 1 month) were not effective. Flurbiprofen (6 months), followed by a combination of flurbiprofen plus alendronate for another 6 months, was effective. Resolution of clinical symptoms was monitored on a scale of one to ten. Improvement of right mandible can be seen in the computed tomography images (arrows).

## 3. Discussion

CNO is a rare, noninfectious, inflammatory bone disease of unknown etiology, which affects mostly children and adolescents (with a female predominance) [1,7]. Previously, Beck et al. looked for mutations in the interleukin-1 receptor antagonist gene in 60 patients with CNO; however, they were not able to confirm that mutations in this gene were an important contributing factor to the pathogenesis of CNO [12]. Additionally, regarding the genetic aspects of CNO, it was found that Human Leukocyte Antigen (HLA)-B27 and HLA-DR -classification in patients with CNO did not differ from those in the general population [13]. CNO was once thought to be one of the most common autoinflammatory bone diseases in childhood [14]. More recently, the definition of bone disease associated with autoinflammatory diseases has expanded to include diseases such as the type I

interferonopathies and bone dysplasia syndromes presenting with hyperostosis-associated systemic inflammation [15]. Host interactions with the microbiome could affect immune homeostasis in inflammatory bowel diseases [16]. As a probable pathogen, *C. acnes* was mentioned in CNO, which was found to be more frequently isolated from open biopsies than percutaneous ones and more frequently positive in patients with simultaneous skin manifestations [5]. Unfortunately, since we did not perform bone biopsy, whether or not *C. acnes* was involved in this case remains unknown.

Clinically, the patient showed no fever besides localized swelling and pain. No night sweats and/or weight loss were noted. It is a challenge to find clinically silent lesions. We think that elevated serum CRP values may reflect active bone lesions, though clinically silent. To find multifocal CNO lesions, PET/CT could be useful. CNO must first be differentiated from BOM, as well as from other rare noninfectious bone diseases such as fibrous dysplasia or juvenile ossifying fibroma. Pediatricians need to be aware of CNO as a diagnosis of exclusion. At one institution, Schnabel et al. [17] reported an almost equal number of pediatric CNO (*n* = 49) and BOM (*n* = 56) cases. The number of bone lesions may help with the diagnosis of differences between CNO and BOM: CNO generally involves multiple (mean = 3) lesions in the metaphysis of the long bones, pelvis, clavicle, and mandible, being more frequent in the clavicle and long bones of children [7,18]. Regarding mandibular lesions, Padwa et al. reported that all 22 CNO cases had multifocal disease [8]. In contrast, Gaal et al. reported an analysis of 22 pediatric CNO mandible cases and concluded that lesions were most often isolated [11]. Considering these contradictory data, careful observation of the clinical course of the involved bone may be required for the differentiation.

Pathologically, Yang et al. showed an osteoid on reactive bone trabeculae, with interstitial granulation tissue, scattered lymphocytic infiltrates, and negative microbiology in the bone of CNO [10]. However, it remains unclear whether a diagnosis of CNO should be based on pathological findings from a bone biopsy [9–11]. In terms of the age of CNO patients at diagnosis, Skrabl-Baumgartner et al. reported a median age of 12.3 years (range, 7.9–18.9) [18]. Another article reported that CNO occurs between the ages of 4 and 14 years, with the average age being 10 years [7]. The fact that CNO is rare in younger children and that the involvement of a single jaw in a child younger than 4 years is likely to be BOM rather than CNO [7] made it difficult for us at first to differentiate CNO from BOM in our patient because she was only 3 years old. A 5-year-old girl with spontaneous jaw pain and large preauricular facial swelling overlying the angle of the mandible, described by Yang et al. [10], was similar to our case, though she was slightly older.

In terms of the diagnosis and follow-up of CNO, imaging studies are essential. MRI is thought to be sensitive for detecting CNO and is considered the gold standard for monitoring the disease [19]. Plain radiograph is difficult for analyzing detailed images of the involved bone. Thus, MRI is more preferable than CT for children considering the risk of X-ray exposure; however, the examination time of MRI is long, which causes a nuisance for younger children, while CT examination can be conducted in a short time. We have employed CT more than MRI, but have attempted to limit CT examinations 6 months apart. On the other hand, based on bone inflammation in CNO, which is associated with increased osteoclastic activity and bone resorption, causing the focal accelerated breakdown of bone collagen, assay of urinary N-terminal telopeptide (NTx) could be helpful as an important biological marker [20,21]. In our patient's follow-up, we have begun testing the usefulness of urinary NTx assay. It is also known that osteopontin (OPN), a secreted phosphoprotein as a member of the small integrin-binding ligand N-linked glycoprotein family of cell matrix proteins, plays a role in bone metabolism and homeostasis; thus, it is closely related to the development of many bone-related diseases [22]. In addition, the dysregulation of the receptor activator of nuclear factor-κB ligand (RANKL) signaling leads to bone diseases such as osteoporosis and osteopetrosis [23]; however, they have not been tested in patients with CNO.

Regarding management and therapeutic responses in the cases of CNO, Gaal et al. summarized drugs employed for 22 mandible CNO cases as NSAIDs (n = 18), glucocorticoids (n = 10), disease-modifying antirheumatic drugs (DMARDs; n = 9), anti-TNF therapy (n = 5), or pamidronate (n = 6) [11]. Full responses were reported in 60% of those receiving anti-TNF therapy and 67% of those receiving pamidronate, which was significantly higher than the full response rate in those taking NSAIDs (11%) ($p < 0.05$). Patients receiving pamidronate responded more rapidly than those receiving anti-TNF therapy (median 2 months vs. 17 months, respectively; $p = 0.01$) [11]. However, in younger children, oral alendronate rather than intravenous pamidronate is preferred. Previously, a 5-year-old girl treated with alendronate (1.5 mg/kg/day, once a week) and vitamin D3 recovered after 3 months [10]. In addition, a 14-year-old girl treated with oral alendronate (35 mg/day, once a week) for 4 months became symptom-free after 2 years [24]. Decisions on the introduction or escalation of treatment must be taken with caution, particularly in young children. In our pediatric CNO case, we started treatment with flurbiprofen for 6 months, which had limited effects (as shown in Figure 2); thereafter, we administered oral alendronate as a combination with flurbiprofen, which was successful.

## 4. Conclusions

CNO, an autoinflammatory, rare, noninfectious, inflammatory bone disease of unknown etiology, needs to be differentiated from BOM. It can be diagnosed with a diagnosis of exclusion, and avoiding the unnecessary administration of antibiotics is possible. CNO should be borne in mind, even in young children aged less than 4 years of age, although the disease mostly affects older children and adolescents. CNO in young children is better treated with oral alendronate rather than NSAIDs.

**Author Contributions:** S.M., H.O., J.S. and S.I. treated the patient. S.M. and S.I. drafted the manuscript. All authors have read and agreed to the published version of the manuscript.

**Funding:** This research received no external funding.

**Institutional Review Board Statement:** The study was conducted in accordance with the Declaration of Helsinki.

**Informed Consent Statement:** Written informed consent was obtained from the patient's parents for publication.

**Data Availability Statement:** No additional data sets are associated with this paper.

**Acknowledgments:** The authors thank Tomoya Masada for his expert opinion on reading images and all medical staffs who took care of the patient.

**Conflicts of Interest:** The authors declare no conflict of interest.

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
