# Peer review of "Chronic Nonbacterial Osteomyelitis of the Jaw in a 3-Year-Old Girl"

_pediatrrep, doi:10.3390/pediatric15010016_

Round 1
Reviewer 1 Report
The case report by Makino et al. presents a 3-year-old gild with chronic non-bacterial osteomyelitis of the jaw. For some reason, the authors have used the template for the MDPI journal “Sustainability”, but should have aimed for “Pediatric Reports” instead.
Nevertheless, the report is fairly well-written and interesting. Below are some specific comments and suggestions for the authors:
· Please report additional information about the hospital she attended (eg., name, state, and country).
· As stated on page 3 “under a probable diagnosis…” is important to emphasize more throughout the manuscript since the authors decided not to perform a biopsy to verify the diagnosis of CNO.
· Figure 1: The 3 images are not perfectly aligned on the horizontal axis. The “C” is quite difficult to see.
· ® should be omitted.
· Figure 2: Please increase the size of the clinical images and provide small white arrows to highlight what changes should be appreciated by the reader.
· Please follow the below statement from the Discussion with additional information about why alendronate is preferred for children: “…However, in younger children, oral bisphosphonate (alendronate) rather than pamidronate is preferred…”
Reviewer 2 Report
Other names have been used for CNO which include SAPHO (Syno-
vitis, Acne, Pustulosis, Hyperostosis, and Osteomyelitis) or nonbacterial
osteomyelitis (NBO). Mention these facts in the introduction section.
Discuss the facts if the patient has cutaneous features, including acne and pustulosis.
In differential diagnosis, systemic autoinflammatory disease with bone involvement, needs to be considered. Please discuss this fact.
Systemic features, such as fever, night sweats and/or weight loss may
occur. The authors need to discuss such.
Key laboratory markers of differential diagnoses such as LDH and uric
acid (malignancies), alkaline phosphatase (hypophosphatasia) and
vitamin C (scurvy) are useful to measure at least initially and remain
normal in CNO. The authors need to highlight such.
Urine N-telopeptide has been suggested to be of value for detecting
flares in CNO patients treated with bisphosphonate. This fact needs to be added.
Which is the preferred imaging tool for monitoring the disease?
How should be clinically silent lesions be screened?
The authors need to discuss cytokine dysregulation and osteoclast activation.
Since certain bacteria alter immune responses, the question of whether pathogens may “indirectly” contribute to disease expression can be raised. These facts need to be discussed.
What are the genetic factors in the pathophysiology of the disease?
Host interactions with the microbiome have been
proposed to affect immune homeostasis and contribute to inflammatory
disease onset. These facts need to be discussed.
Decisions on introduction or escalation of treatment must be taken
with caution. How do the authors address such issues?
Reviewer 3 Report
The article is well written and follows a correct scientific methodology, however I wonder if it was not possible to perform another type of examination, other than a CT scan, on a 3 year old girl. However, I congratulate the authors for the ability to have treated this rare case.
Round 2
Reviewer 2 Report
All necessary corrections were done by the authors.